# Cigarillo Flavor and Motivation to Quit among Co-Users of Cigarillos and Cannabis: A Structural Equation Modeling Approach

**DOI:** 10.3390/ijerph19095727

**Published:** 2022-05-08

**Authors:** Allison M. Glasser, Julianna M. Nemeth, Amanda J. Quisenberry, Abigail B. Shoben, Erika S. Trapl, Elizabeth G. Klein

**Affiliations:** 1Division of Health Behavior and Health Promotion, College of Public Health, The Ohio State University, Columbus, OH 43210, USA; nemeth.37@osu.edu (J.M.N.); klein.232@osu.edu (E.G.K.); 2Department of Health Behavior, Roswell Park Comprehensive Cancer Center, Buffalo, NY 14263, USA; amanda.quisenberry@roswellpark.org; 3Division of Biostatistics, College of Public Health, The Ohio State University, Columbus, OH 43210, USA; shoben.1@osu.edu; 4Department of Population and Quantitative Health Sciences, School of Medicine, Case Western Reserve University, Cleveland, OH 44106, USA; erika.trapl@case.edu

**Keywords:** cigarillos, cannabis, young adults, regulatory science, flavors

## Abstract

Flavored cigar restrictions have the potential to benefit public health. Flavor availability facilitates cigarillo use, but it is unknown if flavor impacts patterns of co-use of cigarillos and cannabis, an increasingly prevalent behavior among young adults. Data were collected (2020–2021) in a cross-sectional online survey administered to a convenience sample of young adults who smoked cigarillos from 15 areas with high cigar use prevalence. We assessed the relationship between flavored cigarillo use and motivation to quit cannabis and cigarillo use among past 30-day co-users (N = 218), as well as several covariates (e.g., cigarillo price and flavor/cannabis policy). Flavored cigarillo perceived appeal and harm were hypothesized parallel mediators. Most co-users reported usually using flavored cigarillos (79.5%), which was not significantly associated with motivation to quit cigarillos or cannabis. Perceived cigarillo harm (β = 0.17, 95% CI = 0.00, 0.33), advertising exposure (β = 0.12, 95% CI = 0.00, 0.24), and income (among racial/ethnic minorities; β = −0.13, 95% CI = −0.25, −0.02) were significant predictors of motivation to quit cigarillos. There were no significant predictors of motivation to quit cannabis. Cigarillo flavor was not associated with motivation to quit, so findings could suggest that banning flavors in cigars may have a neutral impact on co-use with cannabis among young adults.

## 1. Introduction

The leading cause of preventable death and disease is cigarette smoking, with 7 million smoking-attributable deaths occurring each year worldwide [1]. Although cigarette smoking prevalence has declined over time in the United States (US), cigar smoking, which poses similar risk, continues to be popular among young people [2,3]. Cigars have surpassed cigarettes as the most commonly used combustible tobacco product among high-school aged youth [4], and 3.8% of young adults reported smoking cigars every day or some days in 2019 [5]. In the US, cigarillos are disproportionately used by young adults, Black non-Hispanics, males, and those with high school education or less, with an annual household income of <USD 50,000, and who use other tobacco products [6,7]. Black non-Hispanic young adults are nearly four times more likely to smoke cigarillos than White non-Hispanic young adults [6].

Nearly half of cigarillo sales are for flavored products, which are available at 83% of retailers nationwide [8,9]. Flavored products are viewed as more palatable and less harmful and have been shown to increase initiation and progression of tobacco use as well as reduce likelihood of cessation [10,11]. In April 2021, the US Food and Drug Administration (FDA) announced it would propose two product standards to ban menthol as a characterizing flavor in cigarettes and all characterizing flavors in cigars, including menthol [12]. Given that 80% of US youth (12–17 years) and 73% of young adult (18–24 years) tobacco users use a flavored product [13], this proposed regulation could have a significant positive public health impact by reducing initiation and promoting cessation of combustible tobacco products [14,15]. Outside of the US, flavored products are marketed widely, with one tobacco company selling flavored tobacco in over 100 countries in 2015 [16]. As of 2019, flavored tobacco policies have been enacted in 11 countries and the European Union, varying in products covered, flavor types included, and inclusion of restrictions on packaging images/descriptors [16].

New policy or regulation to reduce use of tobacco products needs to account not only for the possible effect on tobacco use, including initiation, cessation, and switching among tobacco products, but also for the potential impact on use of other substances. In particular, cannabis (a plant with psychoactive properties due to tetrahydrocannabinol (THC) content) is commonly co-used with tobacco (use of both products simultaneously or within the same time period, or co-administering them), so should be monitored as tobacco control measures are put in place. The prevalence of cannabis use has increased over time [17], and is highest among young adults [18]. Past-month co-use of cannabis and tobacco (5.4%) is higher among adolescents than use of each product in isolation (cannabis: 2.2%; tobacco: 3.9%) [19]. About three-quarters of adult cannabis users also use tobacco, with co-use highest among young adults [20]. People who co-use may be exposed to higher levels of toxicants than those who use each product alone [21], and co-use is associated with increased frequency of use and dependence on both tobacco and cannabis [22,23]. Co-use has been found to impede cessation of both substances [24], with users reporting similar withdrawal symptoms from both products [25,26,27] and compensation of one product when attempting to quit the other [28]. Additionally, cigar smokers (vs. cigarette smokers) and cannabis users (vs. non-users) in the US are less likely to make an attempt to quit smoking in the past year [29]. Cigars are the most commonly used tobacco product among co-users of tobacco and cannabis [30], but there is a paucity of evidence on the impact of cigar-related policies, particularly flavor policies, on cannabis use.

Three-quarters of cannabis users and two-thirds of blunt (removing tobacco from cigar wrapper and replacing with cannabis) users use cigars [30,31]. Qualitative research suggests that cigarillos are preferred for cannabis use because flavoring masks the smell [32,33] and enhances the experience of creating and smoking blunts [33]. Flavor is among the most highly valued tobacco product attributes for blunt use [34], and one study has reported that 83% of adults who have ever tried a blunt used a flavored cigar wrapper [35]. Flavors in cigar products may influence co-use patterns through altering perceptions of harm and appeal. Flavored products are perceived as less harmful, and lower perceived risk is associated with lower motivation to quit [36,37,38,39,40]. In addition, flavored products are viewed as more appealing than non-flavored products [10] by making the products more palatable and reinforcing, which could make them harder to quit [41,42]. Although these mechanisms (appeal and harm) are supported by evidence for the role of flavor in use of cigars, no studies have determined whether they operate similarly for the role of flavor in co-use of cigars and cannabis, and motivation to quit co-use of these products. In addition, there is a dearth of evidence on cross-substance policy impacts on co-use of cannabis and tobacco.

Research shows that availability of flavors facilitates cigarillo use through the mechanisms of increased appeal and reduced perceived harm of cigarillos [36,37,38,43], but it is unknown whether cigarillo flavor similarly facilitates co-use of cigarillos and cannabis by reducing motivation to quit through these same mechanisms. In this study, our aim was to examine the role of cigarillo flavor, a potential regulatory target, in motivation to quit cigarillos and cannabis among young adults who co-use these products. We hypothesized that among co-users of cannabis and cigarillos, those who use flavored cigarillos will be less motivated to quit cannabis/cigarillo use than those who use unflavored cigarillos, through increased cigarillo appeal and decreased perceived cigarillo harm.

## 2. Materials and Methods

Data are from a non-probability convenience sample of participants in the Cigarillos Flavor and Abuse Liability, Attention, and Substitution (C-FLASH) Study. Young adult cigarillo users were recruited (N = 392), and data were collected in a cross-sectional survey conducted in 2020–2021. The present analysis was limited to cigarillo users who co-used cannabis in the past 30 days (N = 218). All subjects gave their informed consent for inclusion before they participated in the study. The study was conducted in accordance with the Declaration of Helsinki, and the protocol was approved by the Institutional Review Board of Case Western Reserve University (#STUDY20191769).

### 2.1. Study Procedures

Participants were recruited via targeted advertisements posted on social media platforms in 15 geographic regions determined by Youth Risk Behavior Surveillance System data as having higher rates of cigar use [44,45]. Eligibility was determined using a brief, web-based screening survey for individuals: (1) between the ages of 21 and 28 years; (2) smoke an average of two cigarillos per week over the past month, and (3) willing to provide informed consent and participate in the online survey. Participants who screened eligible and provided their email address were sent an email inviting them to provide informed consent and complete a secured online survey, and they received a USD 15 electronic gift card.

### 2.2. Study Measures

#### 2.2.1. Dependent Variables

The dependent variables of interest were motivation to quit cannabis and cigarillos (separately). Participants responded to the following item, assessed separately for each product: “On a scale of 1–10 (1 is least motivated and 10 is most motivated), how motivated are you to quit using the following product: [unflavored/tobacco flavored cigarillos]/[flavored cigarillos]/[marijuana, cannabis, hash, THC, grass, pot, or weed].” Flavored cigarillos were those flavored to taste like fruit, sweets and candy, mint, alcohol, menthol, or some other flavor. All participants were asked to respond to motivation to quit unflavored/tobacco-flavored cigarillos and flavored cigarillos, while only those who reported ever trying cannabis were asked about motivation to quit cannabis. To achieve an overall motivation to quit cigarillo score for those who reported using both unflavored/tobacco- flavored cigarillos and flavored cigarillos, motivation to quit scores for both product types were averaged to maintain a 1–10 scale.

#### 2.2.2. Independent Variables

The primary independent variable of interest was usual flavored cigarillo use (“When you use cigarillos, do you usually use flavors like fruit, sweets and candy, mint, alcohol, menthol or other flavor?”) (yes/no). Based on the Social Contextual Model of Health Behavior [46], a model that takes into account the social context in which health behaviors occur, we examined factors at multiple levels of social ecology (individual, peer, and environmental) to control for the broader context. These covariates included amount usually spent per unit for cigarillos (money typically spent divided by amount typically purchased), annual income, number of people in the household who smoke, frequency of exposure to advertising that promotes smoking in the past six months, use of other tobacco products (cigarettes/e-cigarettes/hookah/smokeless) in the past 30 days, nicotine dependence (scale of 12 proposed items; Appendix A), living in a zip code with a ban on flavored cigarillos (including menthol), and living in a state with legalized recreational cannabis (https://disa.com/map-of-marijuana-legality-by-state, accessed on 11 March 2022).

#### 2.2.3. Mediators and Effect Measure Modifiers

Appeal of flavored cigarillos (7-point Likert scale agreement with, “flavored cigarillos are appealing”) and perceived cigarillo harm (scale of nine proposed items; Appendix A) were hypothesized parallel mediators of the relationship between flavor use and co-use. We examined effect measure modification by gender identity (male vs. female) and race/ethnicity (racial/ethnic minority vs. white).

### 2.3. Data Analysis

For descriptive statistics, we calculated frequencies or means and standard deviations, comparing those who usually use flavored cigarillos and those who do not with Pearson’s Chi-square Test for Independence or Fisher’s Exact Test (for those with cell sizes ≤ 5) [47] for categorical variables, and Student’s *t*-test for continuous variables. For non-normally distributed continuous variables, we calculated median and interquartile range and conducted a two-sample Wilcoxon rank-sum (Mann–Whitney) test for group differences.

We used a structural equation model (SEM) to test the study hypothesis. First, we conducted a confirmatory factor analysis (CFA) to assess measurement of nicotine dependence as a covariate, and harm perceptions as a mediator. As different local and state jurisdictions have varying laws related to tobacco control and legalization of cannabis, we accounted for geographic clustering by state. We conducted multiple-group testing for gender identity and race/ethnicity [10,19,38,48,49,50,51,52,53,54]. All SEM analyses were conducted in Mplus 8 after data preparation in Stata SE 17 and multiple imputation to handle missing data in Blimp Studio 1.3.5. Missing data ranged from 0–8% across variables of interest except the primary outcome variables of interest, which were 14% (motivation to quit cigarillos) and 33% (motivation to quit cannabis) missing. Data were missing at random, and we imputed the data using latent fully conditional specification imputation methods. Fit indices and parameter estimates were averaged across five imputed datasets. We used weighted least square mean and variance adjusted estimation for inclusion of variables measured on an ordinal scale. We present results as standardized parameter estimates.

#### 2.3.1. Power Calculation

After data were collected, we used a test of “not close fit”, where the Root Mean Square Error of Approximation (RMSEA) e_0_ = 0.06 and e_a_ = 0.01, α = 0.05, and a sample size of 218 were used to determine statistical power to conduct these analyses [55]. Using R version 4.0.0 and Preacher’s online calculator [56], we determined that the proposed measurement and structural models of the SEM had power of 0.99 and 0.91, respectively, which is above the desired 0.80 [55].

#### 2.3.2. Model Selection

After finalizing the measurement model based on assessment of fit (Comparative Fit Index (CFI) and Tucker–Lewis Index (TLI) ≥ 0.95 [57], RMSEA estimate ≤ 0.06, and upper 90% confidence limit ≤ 0.06 [55], and standardized root mean squared residual (SRMR) ≤ 0.08 [57]) and any necessary theoretically acceptable modifications, we assessed measurement invariance based on gender identity and race/ethnicity to determine whether statistical models should be run separately [58]. We removed items with non-significant standardized factor loadings of <0.40 or coefficients of determination (R^2^) of <0.50, which may suggest that item is not a good indicator of the latent construct [59]. In the structural models (two separate SEMs were run for motivation to quit cigarillos and motivation to quit cannabis), we removed non-significant correlations from the model, if doing so did not conflict with theory. Alternative models were compared: indirect effects from usual flavored cigarillo use to motivation to quit through perceived harm and appeal (originally hypothesized model; Figure 1), indirect effects through only one pathway (harm or appeal), and no indirect effects.

## 3. Results

### 3.1. Participant Characteristics

Participants (N = 218) were on average 24.6 years of age, and the majority identified as male (56.2%) and reported income of <USD 25,000 per year (58.2%) (Table 1). About one-third each identified as Black non-Hispanic, White non-Hispanic, or Hispanic. Most reported using other tobacco products in the past 30 days (86.8%), usually using flavored cigarillos (79.5%), rarely or sometimes seeing advertisements for smoking (66.5%), and not living in an area with legalized recreational cannabis (71.2%) or a flavored cigarillo ban (90.3%). Participants reported, on average, living with at least one smoker, spending about USD 1.20 on cigarillos, and moderate symptoms of nicotine dependence (14.2 on a scale of 0–32). Mean motivation to quit cigarillos and cannabis scores were 6.8 (SD = 2.9) and 5.6 (SD = 2.4) out of 10, respectively. Compared to participants who usually smoke unflavored/tobacco flavored cigarillos, those who usually smoke flavored cigarillos agreed that flavored cigarillos are appealing, perceived lower cigarillo harm, and reported greater motivation to quit cannabis.

### 3.2. Measurement Model

Results of the CFA confirm a two-factor measurement model with seven indicators of harm perceptions (absolute harm of cigarillos and each type of cigarillo flavor) and eight indicators of nicotine dependence (items related to frequent product use, withdrawal symptoms, persistent use despite negative consequences, and craving) (Table 2). The final model had good fit and was invariant across gender identity and race/ethnicity. Three items hypothesized to load onto perceived harm and four items hypothesized to load onto nicotine dependence were dropped due to low factor loadings (Appendix A). The model explained 59.8% and 59.1% of the total variance in harm and dependence, respectively (both variances were statistically significant at *p* < 0.001), which is considered acceptable in CFA [60].

### 3.3. General SEM

Model fit indices of the hypothesized general SEM did not support an indirect effect of flavored cigarillo use on motivation to quit through cigarillo harm perceptions (for both models measuring motivation to quit each product). Modifying the models to specify only an indirect effect through flavored cigarillo appeal resulted in the best model fit among competing models [57]. However, no findings involving usual use of a flavored cigarillo were statistically significant except that flavored use was a significant predictor of flavor appeal across all models (Table 3; Appendix A).

Perceived harm of cigarillos was significantly positively associated with motivation to quit cigarillos (β = 0.17, 95% CI = 0.00, 0.33), but this effect was not observed when disaggregated by race/ethnicity. More frequent exposure to smoking advertisements in the past six months was positively associated with motivation to quit cigarillos (β = 0.12, 95% CI = 0.00, 0.24). Multiple group testing revealed there were racial/ethnic differences in the magnitude of the parameter estimates for model pathways from perceived harm and flavor appeal to motivation to quit cigarillos, so parameter estimates are presented separately, but the estimates were not statistically significant. Additionally, among racial/ethnic minority young adults, annual income was negatively associated with motivation to quit cigarillos (β = −0.13, 95% CI = −0.25, −0.02). The model was invariant by gender identity.

There were no significant predictors of motivation to quit cannabis use. The model was invariant across categories of gender identity and race/ethnicity. For both models (motivation to quit cannabis and cigarillos), the proportion of variance in motivation to quit explained by the models was low and non-significant (R^2^ < 0.10).

## 4. Discussion

This study was the first to directly assess the role that cigarillo flavor plays in patterns of co-use of cigarillos and cannabis among a sample of young adults, particularly motivation to quit using these products. About 80% of young adult cigarillo and cannabis co-users in this sample usually smoked a flavored cigarillo, which is consistent with other studies finding that 80–90% of cigar/cannabis co-users prefer a flavored cigar [35,61]. Motivation to quit cigarillos was higher than motivation to quit cannabis and it was comparable to quit interest among co-users of tobacco and cannabis in a similar study [28]. Contrary to our hypotheses, usual flavored cigarillo use was not significantly associated with motivation to quit cigarillos or cannabis directly or indirectly through increased appeal of flavored cigarillos. Although usual flavored cigarillo use and perceived appeal of flavored cigarillos were positively associated, as expected [42], there was no relationship between appeal and motivation to quit. Regarding motivation to quit cannabis, some qualitative research suggests that flavors mask the smell of cannabis or enhance the high, so it is possible that flavor is not necessarily appealing for co-users, but has a more utilitarian purpose [32,33,34]. The evidence is fairly robust that menthol in cigarettes may reduce successful cessation of cigarettes [62], but there is limited research on the impact of flavor in quitting or intention to quit cigar products. One mechanism by which menthol has been hypothesized to reduce quitting is by increasing nicotine dependence through increased use [41]. Cigarillos may be used differently than cigarettes, particularly among co-users with cannabis, whose use patterns may differ based on the type of co-use (sequential, co-administration, etc.) [63,64,65,66]. Despite the appeal of flavors, cigarillo use is more social [67,68] and cigarillo smokers tend to smoke less frequently than cigarette smokers [69], so cigarillo use can result in lower nicotine dependence [70,71]; therefore, a lack of physical dependence symptoms may lessen intention to quit.

There was also no effect between usual use of a flavored cigarillo and motivation to quit through perceived cigarillo harm, but there was a positive direct effect of perceived harm on motivation to quit cigarillos. This is consistent with evidence that the greater perceived risk of one’s tobacco use, the greater intention to quit using [39,72]. Interestingly, although not statistically significant, there was an opposing trend for motivation to quit cannabis. Specifically, there was a trend toward lower perceived cigarillo harm being associated with greater motivation to quit cannabis. Studies show that cannabis use can be viewed as less harmful than cigar smoking, and blunt use in particular is seen as less harmful than using cigars with tobacco [37,73]. Therefore, co-users in this study may have been more interested in quitting cigarillos (with tobacco) if cigarillos were viewed as more harmful, but more interested in quitting cannabis if cigarillos were viewed as less harmful, which might suggest they are substitutes. Other evidence is mixed on whether tobacco and cannabis are substitutes or complements [25,26,32,63,74]. More research is needed to understand these complex patterns, particularly to explore motivations to co-use and quit co-using related to relative risk perceptions.

There was also no direct effect of usual flavored cigarillo use on motivation to quit cigarillos or cannabis. Given that most co-users in this sample used flavored cigarillos and other studies suggest that using flavored cigar wrappers for blunts is common [35], flavor may be more of a default option and thus not be a factor of concern when deciding to co-use or quit co-using. Future studies should further explore the importance of flavor in co-use with cannabis as well as the likelihood of quitting each product if flavors were to be banned through a product standard in the US or in other countries or jurisdictions. Evaluation is also needed in countries with different policy environments (e.g., in Canada where flavored cigars are banned and cannabis is legalized) [16].

Greater exposure to past 6-month smoking advertising was found to be associated with greater motivation to quit cigarillos. Research suggests that advertising exposure reduces the likelihood of successfully quitting [75,76]. However, one study found young adults exposed to cigarette advertising were more likely to try using e-cigarettes to quit [77], and another found greater number of quit attempts with exposure to tobacco advertising [78]. This could indicate that those noticing advertisements more often may be more cognizant of their tobacco use, or perhaps they are also being exposed to tobacco warnings, which was not accounted for in this study. Another significant predictor of motivation to quit cigarillos in our study was annual income. Specifically, among racial/ethnic minorities, lower annual income was associated with greater motivation to quit. Among young adults, being unemployed is associated with greater intention to quit [79], and another study found greater intention to quit among Black Americans compared with White Americans [80]. Despite this, low-income and Black smokers have been found to be less likely to quit than their counterparts [81,82]. This could be due to targeted tobacco industry marketing, price promotions, and inadequate access to evidence-based cessation treatment, among other factors [8,83]. Efforts to reduce co-use behaviors may need to consider exposure to marketing in one’s environment and potential targeting of messaging or policy interventions to benefit marginalized populations. In addition, expansion of evidence-based smoking cessation to integrate concurrent cessation of combustible cannabis may also be necessary, as evidence suggests combustible tobacco cessation is less likely among cannabis smokers [84], and co-users who use tobacco routes of cannabis administration are less motivated to reduce tobacco use than those who use non-tobacco routes of administration [85].

Previous analyses of these data found less co-use among people who live in an area with a ban on flavored cigarillos [86], which could suggest removing flavored cigar products could reduce cannabis use. As the vast majority of cigarillos are flavored, a flavor ban may result in reduced availability of cigarillos and blunt wraps, consistent with what was found in an analysis of flavor bans in California [87]. However, living in an area with a flavor ban was not related to motivation to quit cigarillo or cannabis use in this study. A possible explanation could relate to combustible cannabis users switching to other forms of cannabis (e.g., edibles) when flavored cigars are not available versus being motivated to quit all cannabis use. We should note that in our study, a higher proportion of co-users who usually smoked flavored cigarillos lived in areas with a flavored cigarillo ban than those who did not usually smoke flavored cigarillos. Policies are possibly being implemented in areas where a higher proportion of flavored users reside, and/or there may be variability in the strength and enforcement of flavor bans [88].

There were no significant predictors of motivation to quit cannabis use in this study. Given that the proportion of variance in this outcome explained by the model was only 6% (non-significant), there are likely other factors contributing to cannabis use among young adults who use cigarillos that were not assessed in this study.

### Limitations

The proportion of variance explained by the models we ran was <10% and not statistically significant. Other factors, not measured in this study, are likely driving motivations to quit cigarillos and cannabis, and could be targeted in interventions. However, we sought to measure the role of flavor, and assessment of other factors would not likely change the relationship between flavor and motivation to quit in this sample. We did not measure how participants specifically co-use cannabis with cigarillos (i.e., sequentially using, co-administering, using within the same time period but on separate occasions), which may relate differently to the variables we explored in our models. We also did not explore differences by preferred flavor type used, which could reveal heterogeneity of effects. The sampling frame focused on cigarillo users, so our findings may not be able to be generalized to individuals who use other combinations of tobacco and cannabis, such as non-combustible forms of cannabis and tobacco use. Notwithstanding, given that combustible tobacco diminishes success of tobacco cessation, contributes to the most significant health harms, and is disproportionately used by vulnerable populations, we believe this is a critical population of co-users to examine. As a cross-sectional study, we cannot infer causal relationships from this model. However, SEM methodology allowed us to examine indirect effects, multiple relevant pathways simultaneously, and valid measurement of two important variables. Due to power limitations, we combined groups for multiple group testing, which masked possible heterogeneity across racial ethnic minoritized groups. However, we chose to group this way to look for differences between those who are more historically marginalized and may experience the effects of systemic racism in the US versus those who do not to highlight whether policy change (e.g., flavor bans) may be more or less impactful for vulnerable groups.

## 5. Conclusions

Flavor restrictions have great potential to reduce prevalence of tobacco use, especially for cigar smokers, and have begun to show public health impact at the local level in the US [88] and other places globally [89]. Unintended consequences will need to continue to be monitored, including differential impact of restrictions on advantaged vs. disadvantaged groups, responsive shifts in the flavor market by tobacco companies (i.e., flavors invoking cannabis), and impacts on other use of other substances, like cannabis. This study found no relationship between cigarillo flavor use and motivation to quit (cigarillos or cannabis) among co-users of cigarillos and cannabis, suggesting that a possible product standard to ban flavors in cigar products and cigar flavor bans on the local level may have a neutral impact on co-use with cannabis among young adults. This also suggests that for co-using young adults, removal of flavored cigarillos from the market may not deter them from continuing to smoke. However, the growing evidence that flavor restrictions can reduce tobacco use in other populations not limited to co-users [15] indicate that more research is needed to better understand behaviors among the co-using subpopulation, including patterns of and reasons for use. Given the positive relationship between perceived harm of cigarillos and motivation to quit cigarillos in this study, efforts to inform the public on the harms of cigarillos are needed, including communication campaigns and, in the US, reinstating required warning labels on cigar products [90]. Further research is needed to explore the interaction between tobacco and cannabis policy and use of these products.

## Figures and Tables

**Figure 1 ijerph-19-05727-f001:**
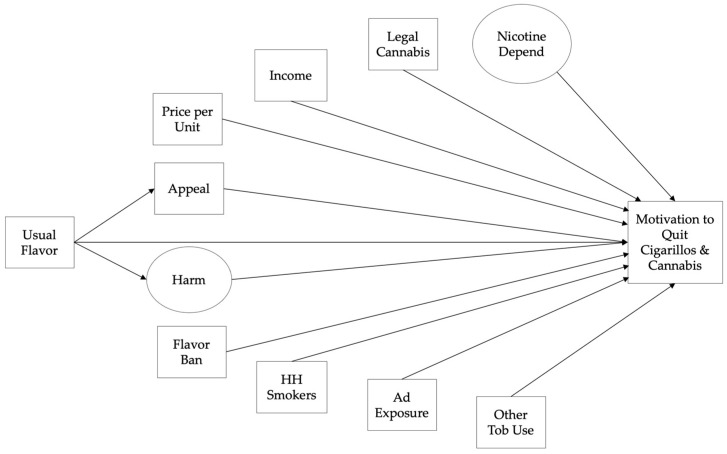
Conceptual Model Diagram (Originally Hypothesized Model): HH = household, Ad = advertising, Tob = tobacco, Depend = dependence.

**Table 1 ijerph-19-05727-t001:** Demographics and Other Characteristics of C-FLASH Young Adults Who Used Cigarillos and Cannabis in the Past 30 Days (N = 218).

			Usually Smoke Flavored Cigarillos ^a^				
	Total (N = 218)		Yes (n = 163; 79.5%)		No (n = 42; 20.5%)		*p* Value
	n	%	n	%	n	%	
Age, mean (SD)	-	24.6 (2.2)	-	24.6 (2.2)	-	24.9 (2.3)	0.342 ^b^
Gender							
Male	122	56.2	96	59.3	23	54.8	0.578 ^c^
Female	91	41.9	62	38.7	19	45.2	
Gender non-conforming	4	1.8	4	2.5	0	0.0	
Race							
Black, non-Hispanic	60	28.8	43	27.2	13	32.5	0.218 ^d^
White, non-Hispanic	62	29.8	45	28.5	15	37.5	
Hispanic	58	25.4	50	31.7	6	15.0	
Other, non-Hispanic	28	13.5	20	12.7	6	15.0	
Annual income							
<USD 25,000	117	58.2	91	59.9	24	58.5	0.983 ^c^
USD 25,000–49,999	55	27.4	40	26.3	12	29.3	
USD 50,000–99,999	25	12.4	18	11.8	5	12.2	
USD 100,000+	4	2.0	3	2.0	0	0.0	
# household smokers, mean (SD)	-	1.7 (0.8)	-	1.8 (0.8)	-	1.6 (0.8)	0.253 ^b^
Other tobacco product use (past 30 days)							
No	30	9.2	21	12.9	9	21.4	0.162 ^d^
Yes	188	86.8	142	87.1	33	78.6	
Price usually paid for cigarillos per unit usually purchased, median (IQR)	-	1.2 (1.0)	-	1.2 (1.0)	-	1.0 (0.7)	0.608 ^e^
Nicotine dependence ^f^, mean (SD, range)	-	14.2 (7.3, 0–32)	-	14.3 (7.5, 0–32)	-	12.1 (6.3, 2–30)	0.080 ^b^
Frequency noticed smoking ads (past 6 months)							
Never	20	9.6	14	8.9	5	12.2	0.086 ^c^
Rarely	65	31.1	46	29.1	18	43.9	
Sometimes	74	35.4	57	36.1	12	29.3	
Often	44	21.1	38	24.1	4	9.8	
Very often	6	2.9	3	1.9	2	4.9	
“Flavored cigarillos are appealing”							
Strongly disagree	8	3.7	2	1.2	6	14.6	**<0.001 ^c^**
Disagree	8	3.7	4	2.5	3	7.3	
Somewhat disagree	14	6.5	6	3.7	6	14.6	
Neither disagree nor agree	25	11.6	14	8.6	9	22.0	
Somewhat agree	46	21.3	38	23.5	6	14.6	
Agree	59	27.3	46	28.4	9	22.0	
Strongly agree	56	25.9	52	32.1	2	4.9	
Perceived cigarillo harm ^f^, median (IQR, range)	-	15.0 (11.0, 0–21)	-	14.0 (11.0, 0–21)	-	21.0 (7.0, 7–21)	**<0.001 ^e^**
Live in area with flavored cigarillo ban							
No	196	90.3	144	88.3	40	95.2	0.151 ^c^
Yes	21	9.7	19	11.7	2	4.8	
Live in area with legalized recreational cannabis							
No	161	71.2	122	74.9	34	81.0	0.408 ^d^
Yes	56	25.8	42	25.2	8	19.1	
Motivation to quit cigarillos, mean (SD) (1–10)	-	6.8 (2.9)	-	6.7 (2.6)	-	6.3 (2.9)	0.517 ^b^
Motivation to quit cannabis, mean (SD) (1–10)	-	5.6 (2.4)	-	5.7 (2.4)	-	4.5 (2.4)	**0.042 ^b^**

^a^ n = 13 missing this item, imputed in SEM analyses; ^b^ Student’s *t*-test; ^c^ Fisher’s Exact Test; ^d^ Pearson’s Chi-square Test for Independence; ^e^ two-sample Wilcoxon rank-sum (Mann–Whitney) test; ^f^ based on confirmatory factor analysis results (see Table 2 for scale indicators); bolded *p* value indicates a statistically significant group difference.

**Table 2 ijerph-19-05727-t002:** Standardized Factor Loadings for Items Assessing Perceived Cigarillo Harm and Nicotine Dependence Among Young Adults Who Used Cigarillos and Cannabis in the Past 30 Days (N = 218).

Construct	Variance	Items ^a^	Factor Loading	R^2^
Cigarillo Harm	59.8%	In general, how harmful do you think cigarillo smoking is to a person’s health?	0.77	0.60
		In general, how harmful do you think smoking a fruit- flavored cigarillo is to a person’s health?	0.91	0.83
		In general, how harmful do you think smoking a sweet and candy-flavored cigarillo is to a person’s health?	0.94	0.88
		In general, how harmful do you think smoking a mint- flavored cigarillo is to a person’s health?	0.92	0.85
		In general, how harmful do you think smoking an alcohol-flavored cigarillo is to a person’s health?	0.89	0.80
		In general, how harmful do you think smoking a menthol-flavored cigarillo is to a person’s health?	0.88	0.77
		In general, how harmful do you think smoking a tobacco-flavored cigarillo is to a person’s health?	0.89	0.79
Nicotine Dependence	59.1%	When I haven’t been able to smoke for a few hours, the craving gets intolerable.	0.77	0.59
		I drop everything to go out and buy tobacco products.	0.79	0.63
		I find myself reaching for tobacco products without thinking about it.	0.72	0.52
		I chain smoke tobacco products.	0.71	0.51
		I feel anxious when I run out of tobacco products.	0.83	0.68
		The only thing that can calm me down is a tobacco product.	0.78	0.61
		I get irritated if I can’t smoke a tobacco product when I feel like using one.	0.73	0.54
		I think about how I will get my next tobacco product.	0.81	0.65

Fit Statistics for Measurement Model (averaged over 5 imputed datasets): RMSEA = 0.045, SD = 0.001; CFI = 0.994, SD = 0.000; TFI = 0.993, SD = 0.000; SRMR = 0.073, SD = 0.000. ^a^ All items had the following response options: (perceived harm) Not at all harmful, Somewhat harmful, Moderately harmful, Very harmful; (nicotine dependence) Never, Rarely, Sometimes, Often, Always.

**Table 3 ijerph-19-05727-t003:** Structural Equation Model Fit Statistics and Standardized Path Estimates.

**Fit Statistic, Mean (SD) ^a^**	**Motivation to Quit Cigarillos**			**Motivation to Quit Cannabis**
RMSEA	0.030 (0.003)	0.026 (0.006)		0.030 (0.003)
CFI	0.991 (0.001)	0.994 (0.002)		0.991 (0.001)
TLI	0.990 (0.002)	0.994 (0.002)		0.990 (0.002)
SRMR	0.074 (0.000)	0.104 (0.000)		0.074 (0.000)
**Pathway, β (95% CI)**	**Overall**	**Racial/Ethnic Minority**	**White**	**Overall**
Usual Flavor (UF)				
UF to Appeal	**0.43 (0.27, 0.59)**	**0.35 (0.07, 0.64)**	**0.83 (0.48, 1.19)**	**0.43 (0.27, 0.58)**
UF to Motivation to Quit (Direct)	0.04 (−0.10, 0.18)	0.16 (−0.07, 0.39)	0.16 (−0.07, 0.40)	0.08 (−0.11, 0.26)
UF to Appeal to Motivation to Quit (Indirect)	−0.01 (−0.08, 0.06)	−0.01 (−0.07, 0.05)	−0.17 (−0.41, 0.07)	−0.02 (−0.08, 0.05)
UF Total	0.03 (−0.10, 0.16)	0.15 (−0.07, 0.38)	−0.01 (−0.24, 0.23)	0.06 (−0.08, 0.20)
Covariates				
Harm to Motivation to Quit	**0.17 (0.00, 0.33)**	0.29 (−0.01, 0.59)	0.10 (−0.20, 0.39)	−0.11 (−0.26, 0.05)
Appeal to Motivation to Quit	−0.02 (−0.19, 0.14)	−0.03 (−0.20, 0.14)	−0.20 (−0.46, 0.06)	−0.04 (−0.19, 0.12)
Per unit price (ln) to Motivation to Quit	0.10 (−0.12, 0.31)	0.13 (−0.07, 0.34)	0.14 (−0.08, 0.35)	−0.02 (−0.22, 0.19)
Dependence to Motivation to Quit	−0.17 (−0.38, 0.05)	−0.17 (−0.38, 0.04)	−0.17 (−0.39, 0.04)	0.11 (−0.07, 0.29)
Income to Motivation to Quit	−0.09 (−0.20, 0.02)	**−0.13 (−0.25, −0.02)**	−0.14 (−0.15, 0.10)	0.01 (−0.17, 0.18)
Household Smokers to Motivation to Quit	−0.03 (−0.16, 0.09)	−0.03 (−0.15, 0.10)	−0.03 (−0.10, 0.34)	−0.04 (−0.14, 0.07)
Ad Exposure to Motivation to Quit	**0.12 (0.00, 0.24)**	0.12 (−0.01, −0.24)	0.12 (−0.01, 0.25)	0.08 (−0.06, 0.21)
Other Tobacco Use to Motivation to Quit	0.07 (−0.05, 0.20)	0.04 (−0.09, 0.17)	0.04 (−0.09, 0.17)	0.02 (−0.14, 0.32)
Flavor Ban to Motivation to Quit	−0.04 (−0.11, 0.03)	0.04 (−0.08, 0.15)	0.04 (−0.08, 0.16)	−0.00 (−0.14, 0.14)
Legal Recreational Motivation to Quit	0.12 (−0.06, 0.30)	0.10 (−0.10, 0.30)	0.10 (−0.10, 0.30)	0.05 (−0.11, 0.20)
**R^2^ for Motivation to Quit**				
	0.05	0.09	0.07	0.06

^a^ Averaged across 5 imputed datasets; to see RMSEA 90% Confidence Intervals for each dataset, see Appendix A. RMSEA = Root Mean Square Error of Approximation; CFI = Comparative Fit Index; TLI = Tucker–Lewis Index; SRMR = standardized root mean squared residual; bold estimates are statistically significant at *p* < 0.05.

## Data Availability

The data presented in this study are available on request from the corresponding author. The data are not publicly available due to ongoing analyses by the study team.

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
