# Peer review of "Cigarillo Flavor and Motivation to Quit among Co-Users of Cigarillos and Cannabis: A Structural Equation Modeling Approach"

_ijerph, 2022, doi:10.3390/ijerph19095727_

Round 1

Reviewer 1 Report

The author suggests banning flavored cigars can have a neutral impact on cannabis co-use, no substantial evidence is obtained from the present study and it is not clear that the authors have collected the type of flavor used in the study participants. Authors may add additional literature support for their claim (lines 321-325). 

Author Response

IJERPH Manuscript 1685350: Response to Reviewers

Reviewer #1

Comment #1: The author suggests banning flavored cigars can have a neutral impact on cannabis co-use, no substantial evidence is obtained from the present study and it is not clear that the authors have collected the type of flavor used in the study participants. Authors may add additional literature support for their claim (lines 321-325). 

Response #1: Thank you for your feedback. The research team did collect the type of flavor used by participants on the survey; however, for this analysis, we were only interested in whether they used any flavor, so do not report specific flavor types in the paper. We have made note of this in the Limitations section, “We also did not explore differences by preferred flavor type used, which could reveal heterogeneity of effects.” Regarding literature support for the claim in lines 321-325, this statement refers to findings from another paper of ours that is under review. We are not aware of any studies, except for the California one (Timberlake, 2021) that we discuss right after that statement, that have explored any interaction with flavor bans and cannabis, so we do not think we can add any further evidence to the text at this point in time.

Reviewer 2 Report

Thank for the opportunity to review this paper, that investigates an original and interesting topic.

Athough the authors' model explains a little portion of variance, the paper opens the way for a new line of research.

The manuscript is not always clear to read, especially I suggest do indicate more clearly the variables analyzed , it not clear the difference between flavored cigarillos and tobacco flavored cigarillos.

I report below some more specific suggestions.

Introduction

Line 72-73: “Nationally in the US, cigar only (vs. cigarette only) and marijuana (vs. no 72 marijuana) users had lower odds of making a quit attempt in the past year.28” It is not clear to me what the authors mean with this. I suggest to rephrase

Overall, the introduction clearly and deeply contextualizes existing literature on the manuscript’s main topic. However, I suggest to be clearer about the main research question, the hypothesis, the aim of the paper and the possible to practical implication of the work, that is why that topic is worth investigation.

Method

Line 123-124: “[unflavored/tobacco flavored cigarillos] / [flavored cigarillos] / [marijuana, cannabis, hash, THC, grass, pot, or weed].” It is not clear to me the difference between “flavored cigarillos” and “unflavored/tobacco flavored cigarillos”. Can the authors describe more precisely this difference for clarity?

The authors calculated through power analysis that a sample size of 218 participants would have consented to an adequate statistical power. 218 participants took part in the research. How could the authors precisely collect the number requested? Did they send invitations progressively? This is not clear. None of participants were excluded? All of them responded correctly to the questionnaire?

Line 210-215: the authors should better describe results. When they say “seven indicators 210 of harm perceptions and eight indicators of nicotine dependence” what are they? Moreover when they say “Three items hy- 212 pothesized to load onto perceived harm and four items hypothesized to load onto nicotine 213 dependence were dropped due to low factor loadings” What are they?

The authors could add a figure representing the final SEM model results.

Discussion

The authors should more deeply underlie the practical implications of the study findings.

Author Response

IJERPH Manuscript 1685350: Response to Reviewers

Reviewer #2

Comment #1: Thank for the opportunity to review this paper, that investigates an original and interesting topic. Although the authors' model explains a little portion of variance, the paper opens the way for a new line of research. The manuscript is not always clear to read, especially I suggest do indicate more clearly the variables analyzed, it not clear the difference between flavored cigarillos and tobacco flavored cigarillos. I report below some more specific suggestions.

Response #1: Thank you for your feedback. We have addressed your concerns in the responses to your comments below.

Comment #2: Introduction / Line 72-73: “Nationally in the US, cigar only (vs. cigarette only) and marijuana (vs. no marijuana) users had lower odds of making a quit attempt in the past year.28” It is not clear to me what the authors mean with this. I suggest to rephrase

Response #2: We have rephrased this statement to make its meaning clearer.

Comment #3: Overall, the introduction clearly and deeply contextualizes existing literature on the manuscript’s main topic. However, I suggest to be clearer about the main research question, the hypothesis, the aim of the paper and the possible to practical implication of the work, that is why that topic is worth investigation.

Response #3: We appreciate this feedback. We edited the final paragraph of the Introduction to more clearly state our research question, study aim, and hypothesis:

“Research shows that availability of flavors facilitates cigarillo use through the mechanisms of increased appeal and reduced perceived harm of cigarillos,35-37,42 but it is unknown whether cigarillo flavor similarly facilitates co-use of cigarillos and cannabis by reducing motivation to quit through these same mechanisms. In this study, our aim was to examine the role of cigarillo flavor, a potential regulatory target, in motivation to quit cigarillos and cannabis among young adults who co-use these products. We hypothesized that among co-users of cannabis and cigarillos, those who use flavored cigarillos will be less motivated to quit cannabis/cigarillo use than those who use unflavored cigarillos, through increased cigarillo appeal and decreased perceived cigarillo harm.”

Comment #4: Method / Line 123-124: “[unflavored/tobacco flavored cigarillos] / [flavored cigarillos] / [marijuana, cannabis, hash, THC, grass, pot, or weed].” It is not clear to me the difference between “flavored cigarillos” and “unflavored/tobacco flavored cigarillos”. Can the authors describe more precisely this difference for clarity?

Response #4: We have added a statement to clarify that flavored cigarillos are those flavored to taste like fruit, sweet and candy, mint, alcohol, menthol, or some other flavor.

Comment #5: The authors calculated through power analysis that a sample size of 218 participants would have consented to an adequate statistical power. 218 participants took part in the research. How could the authors precisely collect the number requested? Did they send invitations progressively? This is not clear. None of participants were excluded? All of them responded correctly to the questionnaire?

Response #5: This analysis was planned as part of a supplement to a larger parent study using data that were already collected (5R01DA048529-03). Therefore, this power calculation was conducted after data collection using the final sample size to determine if we had enough power to run the specific analyses (SEM) that we planned to run. We have added this detail to the sentence, “After data were collected, we used a test of “not close fit”, where the Root Mean Square Error of Approximation (RMSEA) e0=0.06 and ea=0.01, α=0.05, and a sample size of 218 to determine statistical power to conduct these analyses.” We also added the specific power estimates for each model to this section.

Comment #6: Line 210-215: the authors should better describe results. When they say “seven indicators of harm perceptions and eight indicators of nicotine dependence” what are they? Moreover when they say “Three items hypothesized to load onto perceived harm and four items hypothesized to load onto nicotine dependence were dropped due to low factor loadings” What are they?

Response #6: The items included in the measurement model are listed in Table 2 for convenience, but we have parenthetically summarized what they measure. The items dropped are also included in the Supplemental materials, which we had failed to indicate in the text. We have added this information.

Comment #7: The authors could add a figure representing the final SEM model results.

Response #7: We created a figure representing the final SEM model results. Given that there are two models (motivation to quit cigarillos, motivation to quit cannabis), and one of these was stratified by race/ethnicity, the figure is rather busy, and we have decided to put this in the supplemental materials for brevity.

Comment #8: Discussion / The authors should more deeply underlie the practical implications of the study findings.

Response #8: We added additional text to the Conclusions section, providing additional insight into practical implications of this study’s findings.

Reviewer 3 Report

Cigarillo Flavor and Motivation to Quit Among Co-Users of Cigarillos and Cannabis: A Structural Equation Modeling Approach

This study is to determine the role of cigarillo flavor, a potential regulatory target, in co-use of cigarillos and cannabis among young adults . While the topic is of interest, I have some reservations about some aspects of this study.

Abstract

Why is this research vital for the readers?

From where were samples collected? Also, mention the study area.

Which sampling technique was used for data collection?

Add the practical/policy implications.

1. Introduction

Change the reference style in the whole article within the text like [1], [2, 3] and should be hyperlinked with the bibliography.

The authors need to add more literature on developed and developing countries.

Please add the remainder of the study in the last paragraph of the introduction.

2. Materials and Methods

Add regarding sampling technique

Whether dependent and independent variables are based on any theory like the theory of planned behavior or self-developed? Please mention it.

3. Results

Participant characteristics need to explain more according to table 1.

Line 214 - 215: "The model explained 59.8% and 214 59.1% of the variance in harm and dependence, respectively". Why? Explain the reasons.

Table 3: The authors need to explain more in the text for a clear understanding of the readers.

4. Discussion

The study offers an interesting discussion and well organized.

5. Conclusions

What are the practical implications of this study?

Author Response

IJERPH Manuscript 1685350: Response to Reviewers

Reviewer #3

Comment #1: This study is to determine the role of cigarillo flavor, a potential regulatory target, in co-use of cigarillos and cannabis among young adults. While the topic is of interest, I have some reservations about some aspects of this study.

Response #1: Thank you for your feedback. We have addressed your comments in our responses below.

Comment #2: Abstract / Why is this research vital for the readers?

Response #2: The abstract word count is limited, so we have tried our best to address this comment and the next three related to the abstract, while staying within the word limit. For this comment, we have added the statement, “Flavored cigar restrictions have the potential to benefit public health.” to underline the importance of exploring this topic.

Comment #3: Abstract / From where were samples collected? Also, mention the study area.

Response #3: We have added that the participants came from 15 geographic areas with high known cigar use prevalence.

Comment #4: Abstract / Which sampling technique was used for data collection?

Response #4: We have added that this was a convenience sample.

Comment #5: Abstract / Add the practical/policy implications.

Response #5: In the abstract, we have included the following statement regarding policy impacts, “Cigarillo flavor was not associated with motivation to quit, so findings could suggest that banning flavors in cigars may have a neutral impact on co-use with cannabis among young adults.”

Comment #6: Introduction / Change the reference style in the whole article within the text like [1], [2, 3] and should be hyperlinked with the bibliography.

Response #6: We have updated the reference style.

Comment #7: The authors need to add more literature on developed and developing countries.

Response #7: We have added some global context in the Introduction, “Outside of the US, flavored products are marketed widely, with one tobacco company selling flavored tobacco in over 100 countries in 2015.[16] As of 2019, flavored tobacco policies have been enacted in 11 countries and the European Union, varying in products covered, flavor types included, and inclusion of restrictions on packaging images/descriptors.[16]”

Comment #8: Please add the remainder of the study in the last paragraph of the introduction.

Response #8: We edited the final paragraph of the Introduction to more clearly state our research question, study aim, and hypothesis:

“Research shows that availability of flavors facilitates cigarillo use through the mechanisms of increased appeal and reduced perceived harm of cigarillos,35-37,42 but it is unknown whether cigarillo flavor similarly facilitates co-use of cigarillos and cannabis by reducing motivation to quit through these same mechanisms. In this study, our aim was to examine the role of cigarillo flavor, a potential regulatory target, in motivation to quit cigarillos and cannabis among young adults who co-use these products. We hypothesized that among co-users of cannabis and cigarillos, those who use flavored cigarillos will be less motivated to quit cannabis/cigarillo use than those who use unflavored cigarillos, through increased cigarillo appeal and decreased perceived cigarillo harm.”

If this is not what was intended by this reviewer’s comment, please let us know.

Comment #9: Materials and Methods / Add regarding sampling technique

Response #9: We have added this detail to the Methods section: “Data are from a non-probability convenience sample of participants in the Cigarillos Flavor and Abuse Liability, Attention, and Substitution (C-FLASH) Study.”

Comment #10: Whether dependent and independent variables are based on any theory like the theory of planned behavior or self-developed? Please mention it.

Response #10: We added the following to the Methods section relevant to measures: “Based on the Social Contextual Model of Health Behavior,[46] a model that takes into account the social context in which health behaviors occur, we examined factors at multiple levels of social ecology (individual, peer, and environmental) to control for the broader context. These covariates included […]”

Comment #11: Results / Participant characteristics need to explain more according to table 1.

Response #11: We have added to the text those characteristics from Table 1 previously not described.

Comment #12: Line 214 - 215: "The model explained 59.8% and 59.1% of the variance in harm and dependence, respectively". Why? Explain the reasons.

Response #12: This is the total variance explained by the indicators in the model. I have provided further information that these values were statistically significant and that these are acceptable values in the social sciences: “The model explained 59.8% and 59.1% of the total variance in harm and dependence, respectively (both variances were statistically significant at p<0.001), which is considered acceptable in CFA.[60]”

Comment #13: Table 3: The authors need to explain more in the text for a clear understanding of the readers.

Response #13: In this revised version, we have added new text and made language modifications throughout the results section to provide clarification of the key findings.

Comment #14: Discussion / The study offers an interesting discussion and well organized.

Response #14: Thank you for this feedback.

Comment #15: Conclusions / What are the practical implications of this study?

Response #15: We have updated the Conclusions paragraph to more clearly state what the practical implications of the study are.

Reviewer 4 Report

You all did a pretty journeyman-like job on the design and analysis.  Given the modest sample, and the covariates involved, statistical significance was not too informative, but you made some sound points that should inform public health policy in the near future.

Author Response

IJERPH Manuscript 1685350: Response to Reviewers

Reviewer #4

Comment #1: You all did a pretty journeyman-like job on the design and analysis.  Given the modest sample, and the covariates involved, statistical significance was not too informative, but you made some sound points that should inform public health policy in the near future.

Response #1: Thank you for your feedback!

Round 2

Reviewer 2 Report

The paper is now suitable for publication

Reviewer 3 Report

The authors have incorporated all comments.